# Use of Tuna Visceral Pepsin in Combination with Trypsin as Digestion Aid: Enhanced Protein Hydrolysis and Bioavailability

**DOI:** 10.3390/foods12010125

**Published:** 2022-12-26

**Authors:** Umesh Patil, Jirakrit Saetang, Bin Zhang, Soottawat Benjakul

**Affiliations:** 1International Center of Excellence in Seafood Science and Innovation, Faculty of Agro-Industry, Prince of Songkla University, Hat Yai, Songkhla 90110, Thailand; 2Key Laboratory of Health Risk Factors for Seafood of Zhejiang Province, College of Food Science and Pharmacy, Zhejiang Ocean University, Zhoushan 316022, China

**Keywords:** fish enzymes, Caco-2 cell monolayers, protein digestion, antioxidant activity

## Abstract

Freeze-dried tuna pepsin powder (TPP) was prepared using maltodextrin (10%) and trehalose (5%), while trypsin-loaded beads (TLB) with 5% glycerol were obtained via chitosan/alginate ionotropic gelation. The storage stability of TPP and TLB and their proteolytic activity toward red kidney bean protein (RKB), threadfin bream surimi (TBS) and egg white protein (EWP) in varying simulated *in vitro* gastrointestinal (GI) tract conditions were studied. The intestinal transepithelial transportation of generated peptides was also carried out through Caco-2 cell monolayers after the cytotoxicity test. Enzyme activity was dropped when TPP and TLB in blister packs were kept for 10 weeks of storage at room (28 °C) and refrigerated (4 °C) temperatures. TPP and TLB at a level of 50% (*w*/*w* of proteins) effectively hydrolyzed RKB, TBS and EWP in a simulated *in vitro* GI tract, as indicated by marked protein degradation and increased degree of hydrolysis. Some peptides generated after GI digestion could transport through Caco-2 cell monolayers. Those peptides had different molecular size distribution and antioxidant activities. The highest antioxidant activity was observed for RKB hydrolysate after passing through the Caco-2 cell monolayer. Therefore, TPP and TLB from skipjack tuna viscera could potentially be used for enzyme supplementation to help digest food proteins. Food-derived bioactive peptides generated after GI digestion could assist in improving human health due to their antioxidant activity.

## 1. Introduction

Digestive enzymes are necessary to digest foods, particularly proteins. In order to cleave proteins into amino acids (AAs) and smaller peptides before absorption in the small intestine, pepsin first partially hydrolyzes proteins, followed by trypsin hydrolysis [1]. However, the digestive system is affected by modern lifestyles and aging [2]. The nutritional availability and functional potency of food-derived peptides are mainly dependent on the hydrolysis of protein-rich foods. Maldigestion could be caused by a lack of digestive enzymes, particularly in elderly people or persons with several digestive diseases [3]. Therefore, digestive enzyme supplementation is an alternative for individuals with enzyme insufficiencies. Commercial production of pepsin and trypsin supplementation are obtained from porcine or bovine sources [4,5]. Nevertheless, religious limitations for consumption are of the major concern. Patil et al. [6] demonstrated that fish enzymes, including pepsin and trypsin from skipjack tuna viscera, can be utilized as a potential alternative for enzyme supplementation. Freeze-dried tuna pepsin powder (TPP) was prepared with the aid of maltodextrin (10%) and trehalose (5%). Trypsin-loaded beads (TLB) with 5% glycerol were produced by entrapping the enzyme in chitosan/alginate gel [7]. TPP had efficacy in hydrolysis of proteins in a simulated *in vitro* gastric digestion system, and TLB demonstrated their efficient delivery capability, in which trypsin activity was retained in gastric condition. Thereafter, dissociation and release of trypsin occurred, and protein hydrolysis was consequently augmented in the intestinal tract.

The simulated *in vitro* GI digestion method is generally used to mimic the digestion process. Moreover, the Caco-2 cell monolayer has been widely used as an *in vitro* model for intestinal absorption across the epithelium. Although this type of cell is a human colorectal cancer cell line, it displays an enterocyte-like phenotype after culturing as a monolayer for 20–22 days [8]. Many types of enzymes and transport molecules expressed in the human small intestine were also found in fully differentiated Caco-2 cell monolayers [9]. Moreover, the previous study demonstrated the correlation between this model and in vivo human absorption for a wide range of drug molecules [10]. Recently, the Caco-2 cell monolayer has been used for assessing the intestinal permeability of peptides or nutraceuticals [11]. Peptides and AAs generated after protein hydrolysis can be rapidly absorbed by the body, thus boosting nutrient delivery to muscle tissues [6]. Apart from their nutritional value, bioactive peptides have excellent abilities for improving human health, for instance, as antioxidant, antihypertensive agent, etc. [12]. AAs and peptides are mainly absorbed by epithelium cells of the small intestine. Previous findings showed the antioxidant properties of several food protein hydrolysates, including those from fish [13], egg [14] and milk [15], etc. Peptides sequences, AA content, and molecular weight distribution generally have a major impact on antioxidant properties of hydrolysates [16]. Enzymes present in the GI tract can induce proteolysis [17]. Moreover, enzymes from different sources with unique specificity towards substrate also determine the characteristics of generated peptides [6]. Fish enzymes can hydrolyze food proteins effectively in the absence of digestive enzymes [6]. This could help produce food-derived peptides with the ease for absorption. Furthermore, the generated peptides should remain active after being transported across the Caco-2 cell monolayer. Currently, information on the Caco-2 cell monolayer absorption and transportation of digested proteins by fish digestive enzymes is limited. Therefore, the objectives of the present study were to investigate the storage stability of TPP and TLB and to investigate their hydrolytic activity toward red kidney bean protein (RKB), threadfin bream surimi (TBS) and egg white protein (EWP) under varying simulated *in vitro* GI tract conditions. Additionally, molecular weight distribution and antioxidant activities of generated peptides after Caco-2 cell monolayer transportation were also studied.

## 2. Materials and Methods

### 2.1. Chemicals and Materials

All the chemicals used were of analytical grade and purchased from Sigma (St. Louis, MO, USA). The pH shift method was used to prepare red kidney bean protein isolate (RKB), as tailored by Gulzar et al. [18]. Threadfin bream surimi (TBS) (Grade B) was bought from Chaichareon Marine Co., Ltd. (Pattani, Thailand). Egg white protein (EWP) was acquired from Cottage farm (Bangkok, Thailand).

### 2.2. Preparation of Skipjack Tuna Pepsin Powder

Freeze-dried skipjack tuna pepsin powder containing maltodextrin (10%) and trehalose (5%) was prepared as tailored by Patil, Nikoo, Zhang and Benjakul [6]. Briefly, a crude pepsin extract solution was used to dissolve maltodextrin (10%, *w*/*v*) and glycerol (5%, *w*/*v*). The mixture was then freeze-dried using a SCANVAC CoolSafe™ freeze-dryer (CoolSafe 55, ScanLaf A/S, Lynge, Denmark). The obtained powder was labeled “tuna pepsin powder” (TPP), packaged in zip lock bag and stored at −40 °C until further analyses. The proteolytic activity of TPP was assessed using hemoglobin as a substrate following the method of Patil, Nikoo, Zhang and Benjakul [6].

### 2.3. Preparation of Skipjack Tuna Trypsin-Loaded Beads

Trypsin-loaded beads containing 5% glycerol were prepared using chitosan and alginate ionotropic gelation as guided by Patil, Nagarajarao, Balange, Zhang and Benjakul [7]. Two grams of crude trypsin freeze-dried powder was added to 100 mL of alginate solution (2%, *w*/*v* in distilled water, DI) containing glycerol (5%, *w*/*v*). The mixture was stirred at 250 rpm using a magnetic stirrer. A completely dissolved mixture was dripped into chitosan (LWM) solution (2% *w*/*v*, pH 5.5, containing 10% CaCl_2_, *w*/*v*) with the aid of a peristaltic pump (2 mL/min) connected to a standard pipette tip (0.46 mm diameter). The resulting beads were rinsed twice with 10 volumes of DI and freeze-dried. The freeze-dried beads were named as “trypsin loaded beads” (TLB), packed in zip-lock bags and stored at −40 °C until further analyses. The trypsin activity of crude trypsin power was assayed using Nα-Benzoyl-DL-arginine p-nitroanilide hydrochloride (BAPNA) as a substrate [7].

### 2.4. Storage Stability of TPP and TLB Capsules in Blister Pack at Various Temperatures

During the extended storage, the enzymes could undergo the degradation or autolysis associated with the loss of enzyme activity [6]. Therefore, TPP and TLB were filled into hard capsules separately and packed in blister packs to mimic the commercial products, which are mainly packaged in hard capsules (Figure 1). TPP and TLB hard capsules were stored at refrigerated (4 °C) and room temperature (28 °C) for up to 10 weeks. To measure the relative activity during the storage, samples were randomly taken every week. The relative pepsin and trypsin activities of hard capsules containing TPP and TLB were evaluated as tailored by Patil, Nikoo, Zhang and Benjakul [6] and Patil, Nagarajarao, Balange, Zhang and Benjakul [7], respectively.

The relative pepsin or trypsin activity was calculated as follows:$$Relative\;activity\;\left(\%\right)=\frac{S_{f}}{S_{i}}\times 100$$
where *S_f_* is TPP or TLB proteolytic activity after designated storage. *S_i_* is initial TPP or TLB proteolytic activity at day 0.

### 2.5. Simulated In Vitro Gastrointestinal (GI) Digestion of Some Proteins by TPP and TLB

Simulated *in vitro* GI digestion of RKB, TBS and EWP was performed as described by Patil, Nagarajarao, Balange, Zhang and Benjakul [7]. TPP and TLB were employed in a simulated *in vitro* GI tract instead of commercial enzymes. Digestion was performed under various conditions, involving (a) complete GI digestion, in which TPP and TLB were added; (b) inactivated gastric digestion and only TLB was added; (c) inactivated intestinal digestion and only TPP was added; and (d) a control sample, without TPP and TLB added (Figure 2). Protein samples (2 g) were transferred into 10 mL of 0.1 N HCl and pH 2.0 was adjusted by 0.1 M HCl. TPP and/or TLB at amounts of 25% and 50% (w/w of protein in the sample) were added with the mixture to initiate hydrolysis. The pH was checked and readjusted to 2.0 using 0.1 M HCl. In order to mimic *in vitro* gastric digestion, each sample was incubated in an orbital shaking water bath (Memmert, D-91126, Schwabach, Germany) at 37 °C for 1 h. The pH of gastric digest was then increased to 5.3 by the addition of sodium bicarbonate (0.9 M). Freshly prepared bile salts (200 μL) were subsequently added. Then, the pH was raised to 7.4 using 1 M NaOH. Samples were incubated at 37 °C for another 2 h with constant shaking to mimic the duodenal phase. The digest was submerged in boiling water for 10 min to terminate the hydrolysis reaction. The digest was then cooled and centrifuged at 8500× *g* for 15 min (Beckman Coulter, Allegra™ centrifuge, Palo Alto, CA, USA). The supernatant was collected and kept at 4 °C until further analysis.

#### 2.5.1. Degree of Hydrolysis (DH) of Digests

The DH of the samples was measured as detailed by Patil and Benjakul [19]. Briefly, 125 μL of diluted sample was mixed with 2.0 mL of 0.2 M phosphate buffer, pH 8.2 and 1.0 mL of TNBS (0.01%, *w*/*v*) solution. The mixture was vortexed and incubated for 30 min at 50 °C in the dark. To terminate the reaction, 2.0 mL of 0.1 M sodium sulfite were added to the reaction mixture. The mixtures were subsequently cooled at room temperature for 15 min. The absorbance was read at 420 nm using a UV-1601 spectrophotometer (Shimadzu, Kyoto, Japan) and α-amino group was expressed in terms of L-leucine. The DH was calculated as follows:$$\mathrm{DH}\left(\%\right)=\frac{L-L_{0}}{L_{max-}L_{0}}\times 100$$
where *L* is the amount of α-amino groups of a hydrolyzed sample. *L*_0_ is the amount of α-amino groups in the initial sample. *L_max_* is the total α-amino groups obtained after acid hydrolysis (6 M HCl at 100 °C for 24 h).

#### 2.5.2. SDS-PAGE of Digests

SDS-PAGE was used to analyze the protein patterns of the digested protein samples [20]. The sample was diluted with SDS (5%) (1:1, *v*/*v*), heated at 95 °C for 30 min and centrifuged for 15 min at 8000× *g*. The Biuret method was used to measure protein content of the solution [21]. The protein (15 μg) was loaded on freshly prepared polyacrylamide gel (12% running gel; 4% stacking gel). The gels were stained and destained after separation at 15 mA per gel. The samples showing the highest degradation of proteins were selected for further experiments.

### 2.6. Measurement of Cytotoxicity

MTT assay was adopted to determine the cytotoxicity of hydrolysate prepared after GI digestion in Caco-2 cells [22]. Caco-2 cells were cultured in 96-well culture plates and incubated with different concentrations of hydrolysate (1 and 5 mg/mL), which were dissolved in DMEM (Dulbecco’s Modified Eagle Medium). Incubation was performed at 37 °C in 5% CO_2_ for 4 h and 24 h, followed by the addition of MTT (0.05 mg) and incubation for additional 4 h. The culture medium was then substituted with dimethyl sulfoxide (DMSO) (150 μL) to solubilize the MTT crystals. The absorbance at 595 nm was measured using a FLUOstar^®^ Omega microplate reader (BMG Labtech, Ortenberg, Germany). The cell viability of Caco-2 cells was calculated based on the absorbance, and the cytotoxic effect was evaluated by their cell viability.

### 2.7. Transport Studies

#### 2.7.1. Cell Culture

Caco-2 cells were obtained from human colon adenocarcinomas and cultured in DMEM containing antibiotic–antimycotic (1%, *v*/*v*) and fetal bovine serum (10%, *v*/*v*) at 37 °C under 5% CO_2_ in a fully humidified atmosphere. After cells (10–12 passages) reached a confluency of 80–90%, the cells were detached using 0.25% trypsin–EDTA and sub-cultured. For transport experiments, Caco-2 cells (1.2 × 10^5^ cells/cm^2^) were inoculated onto 12-well Transwell inserts (0.4 μm pore size, 12 mm diameter; Corning Costar, Corning, NY, USA) and grown for 25–31 days to fully differentiate the cells. Every other day, the medium was removed and replaced with freshly prepared medium. The integrity of cell monolayers was measured using a Millicell ERS device (Millipore, Billerica, MA, USA). The actual transepithelial electrical resistance (TEER) values were computed by subtracting the ER of cell culture inserts without cells. For the transportation study, a Caco-2 cell monolayer with TEER values over 350 Ω cm^2^ was employed.

#### 2.7.2. Apical-to-Basolateral Transport Studies

Prior to the transport study, the Caco-2 cell monolayer was rinsed twice with Hanks’ Balanced Salt Solution (HBSS) to remove the remaining culture media and further equilibrated with HBSS for 30 min at 37 °C. The medium was removed and replaced with 0.5 mL of hydrolysate dissolved in HBSS (1 and 5 mg/mL) on the apical (AP) side and 1.5 mL of fresh HBSS on the basolateral (BL) side. The blank was prepared in the same way without the addition of sample. The cell monolayer was allowed to incubate at 37 °C with 5% CO_2_ for 1 h, 2 h and 4 h. Subsequently, the permeates on the BL-side were collected and used for further analysis.

##### Percentage of Permeability

The sample on the AP-side and their corresponding Caco-2 cells permeate from the BL compartment at various times (1 h, 2 h and 4 h) was measured for peptide concentration, as reported by Samaranayaka, Kitts and Li-Chan [13]. TNBS method was used to determine peptide concentration by measuring the α-amino group content. TNBS could react with N-terminal amino groups to form trinitrophenyl-amino acid derivatives, which were then determined by spectrophotometer [23]. In Brief, 125 μL of sample was added to 3 mL mixture of phosphate buffer (0.2 M, pH 8.2) and TNBS (0.01%, *w*/*v*) at a ratio of 2:1, (*v*/*v*). The mixture was vortexed and incubated for 30 min at 50 °C in the dark. To stop the reaction, 2.0 mL of 0.1 M sodium sulfite was added. The absorbance was read at 420 nm and L-leucine (0.5 to 5.0 mM) was used as the standard.

The efficacy of peptide transportation was expressed as the percentage of permeated peptides and calculated using the following equation:$$Permeation\;\left(\%\right)=\left(\frac{P_{basolateral}-P_{blank}}{P_{apical}}\right)\times 100$$
where *P_apical_* and *P_basolateral_* are the peptide amounts on the AP-side before transport and that found in BL-side after designated time of transport, respectively. *P_blank_* is the peptide amounts on the BL-side of the blank after designated transport time.

The samples having the highest percentage of permeation were selected for further experiments.

##### Size Distribution of Digests

The selected sample on the AP-side and their corresponding Caco-2 cells permeates on the BL-side were investigated by MALDI-TOF as described by Benjakul et al. [24]. The Autoflex Speed MALDI-TOF (Bruker, GmbH, Bremen, Germany) mass spectrometer equipped with a 337 nm nitrogen laser was used.

##### Antioxidant Assay

The samples on the AP-side and their corresponding Caco-2 cells permeates on the BL-side were used for antioxidant assay. DPPH and ABTS radical scavenging activities were assayed as per the methods detailed by Mittal et al. [25]. Units were expressed as mmol Trolox equivalent (TE)/L of digest.

### 2.8. Statistical Analysis

A completely randomized design (CRD) was applied for the entire study. The experiments were conducted in triplicates using three lots of samples. Each analysis was also conducted in triplicate. The data was subjected to an analysis of variance (ANOVA). The mean comparison was done using Duncan’s multiple range test. The t-test was performed to compare the pairs [26]. The Statistical Package for Social Science (SPSS 22.0 for Windows, SPSS Inc., Chicago, IL, USA) was used to conduct the statistical analysis.

## 3. Results and Discussion

### 3.1. Storage Stability of TPP and TLB Capsules in Blister Pack at Various Temperatures

The storage stability of TPP and TLB capsules in blister packs at various temperatures (4 °C and 28 °C) is displayed in Figure 3. The enzyme activities of TPP and TLB were consistently decreased over time, regardless of temperature. The loss in a relative activity of TPP and TLB at both temperatures indicated that pepsin and trypsin plausibly underwent denaturation during extended storage. At week 1, the highest relative proteolytic activity was observed for both samples stored at 4 °C, while lowest activity was found at week 10 (*p* < 0.05). Samples kept at 4 °C showed higher relative activity than those stored at 28 °C, particularly for extended storage time (10 weeks). Similar results were reported by Buamard, Aluko and Benjakul [3], in which beads loaded with trypsin showed higher enzymatic stability at 4 °C than room temperature. Lower temperatures caused a decrease of molecular alteration [27]. Therefore, denaturation of enzymes could be retarded. A loss of enzyme activity of TPP by 38.78% and 48.30% was found after storage for 10 weeks at 4 °C and 28 °C, respectively. Similarly, enzyme activity for TLB was decreased by 33.75% and 44.59% at 4 °C and 28 °C, respectively. TLB showed less decrease of enzyme activity than TPP, particularly at week 10. TPP was prepared using maltodextrin (10%) and trehalose (5%). However, TLB was entrapped in bead in the presence of 5% glycerol. Thus, the encapsulating materials used in TLB played a crucial role in enzyme stabilization, particularly during longer storage time. To enhance the stability of TPP and TLB placed in hard capsules and further blister packed, a dry condition with a low temperature was recommended.

### 3.2. Simulated In Vitro Gastrointestinal (GI) Digestion of Some Proteins by TPP and TLB

#### 3.2.1. Degree of Hydrolysis (DH)

The DH of protein samples, including RKB, TBS and EWP with varying simulated *in vitro* GI tract conditions without and with TPP and/or TLB at various levels (25 and 50% of proteins), is presented in Table 1. Different DHs were noted for each sample, depending on the GI tract conditions and levels of enzymes applied. The results suggested that the number of peptide bonds were cleaved in all protein samples at different levels. For the RKB sample, the DH increased with higher levels of enzymes used in all GI tract conditions (*p* < 0.05). The addition of enzymes, especially at high levels, caused greater protein hydrolysis. The DH was lower in the absence of TPP, suggesting that TPP played an excellent role in protein hydrolysis. The results indicated that the RKB sample was more susceptible to hydrolysis by TPP than TLB. The highest DH was noted in the presence of both digestive enzymes in the GI tract, particularly when both enzymes at 50% were used. The results indicated that the combined digestive effects of TPP and TLB in the simulated *in vitro* GI condition augmented the degradation of protein at a higher level.

For the TBS sample, a similar DH result was observed in all GI tract conditions and enzyme levels used. However, the higher DH was observed in the TBS sample when compared to the RKB and EWP samples, especially when higher levels of enzymes were used (50%) (*p* < 0.05). The TBS sample was, therefore, more susceptible to hydrolysis than RKB and EWP samples by enzymes used. The addition of TPP and TLB, especially at higher levels, increased the hydrolysis of proteins, as evidenced by the higher DH. It is well known that fish proteins are easily digested [28] and those proteins are suitable for people with digestive abnormalities. Buamard, Aluko and Benjakul [3] documented similar results, in which trypsin entrapped in chitosan-alginate beads hydrolyzed sardine fish mince at a greater level than sodium caseinate and soy protein isolate. The EWP sample also showed similar DH results as the RKB and TBS samples. Nevertheless, different DHs were attained. Thus, skipjack tuna enzymes, both pepsin and trypsin, could be used as an alternative enzyme supplement for elderly or person with enzyme depletion. Moreover, it has no religious constraints like commercially available porcine or bovine pepsin and trypsin.

#### 3.2.2. Protein Patterns

Protein patterns of the RKB, TBS and EWP samples after digestion by various simulated *in vitro* GI tract conditions without and with TPP and/or TLB addition at varying levels (25 and 50% of proteins) are displayed in Figure 4. The RKB sample (undigested sample) having several protein bands, including molecular weight (MW) of 55 kDa, 35 kDa, 29 kDa, 19 kDa and a band with MW above 55 kDa (Figure 4A, Lane U). Upon enzymatic hydrolysis, those protein bands vanished or were hydrolyzed with the concomitant production of peptides or proteins with low MW. Thus, protein degradation was most probably caused by TPP and TLB. More degradation of proteins was noticeable as the enzyme level increased, particularly at 50% (Figure 4A, Lane 25 and 50). The protein pattern of the RKB sample was slightly changed when TPP and TLB were used alone for hydrolysis. A protein band with 97 kDa MW appeared after the RKB sample was hydrolyzed under gastric inactivation and complete GI digestion condition, but this phenomenon was not found under intestine inactivation condition. Most protein bands almost disappeared under complete GI digestion condition at 50% enzymes. Pepsin is an endopeptidase that cleaves aromatic AAs (such as tyrosine and phenylalanine) from the N–terminus of proteins [29]. Trypsin is serine protease that is very specific for the cleavage of peptide bonds consisting of the carboxyl groups of lysine and arginine residues [19]. In general, pepsin and trypsin have unique cleavage sites that probably produce peptides with varying MWs. A protein with MW of 55 kDa was retained, irrespective of simulated *in vitro* GI conditions and enzyme concentration. This was most likely because those plant proteins were highly resistant toward fish enzymes (TPP and TLB). Plant proteins might contain protease inhibitors or phytochemicals, especially phenolic compounds, which plausibly inactivate enzymes via cross-linking action [30]. Thus, protein hydrolysis became lowered.

On the other hand, obvious hydrolysis was noted for the TBS sample after hydrolysis by TPP and TLB, especially at 50% (Figure 4B, Lanes 50). TBS (undigested sample) had distinctive protein bands of tropomyosin (TM), actin (AC), and myosin heavy chain (MHC) [31] (Figure 4B, Lane U). However, those protein bands were totally degraded or vanished after being hydrolyzed by TPP and TLB with concomitant appearance of peptides with low MW. The results revealed that both enzymes efficiently hydrolyzed the TBS samples and yielded different protein patterns. The results were in agreement with DH, where the highest DH was noticed for the TBS sample, indicating the drastic protein hydrolysis. The protein patterns of the EWP (undigested sample) (Figure 4C, Lane 2) showed the typical bands for proteins of egg white, which are lysozyme, ovomucoid, ovalbumin and ovotransferin. These protein bands were in accordance with the results documented by Kuan et al. [32]. Additionally, a band over 97 kDa MW was observed. Commercial EWP used in baking plausibly contained the cross-linked proteins with high MW. Ovotransferrin and lysozyme bands were completely degraded in all GI condition and enzyme levels used. Ovalbumin and ovomucoid were partially hydrolyzed when TPP and TLB were used alone. Nevertheless, these protein bands disappeared when complete GI condition (both TPP and TLB), especially at a higher level of enzymes (50%) were used. In general, protein hydrolysates generated after proteolysis mainly contained AAs and low MW peptides, both of which possibly have nutritional or nutraceutical advantages. The results revealed that pepsin and trypsin from skipjack tuna viscera could potentially help digest all food proteins to a high level. As a result, those two proteases could assist digestion in patients with limited secretion of digestive enzymes.

### 3.3. Cell Cytotoxicity

The toxicity of hydrolysates generated after simulated *in vitro* GI digestion toward Caco-2 cells is shown in Figure 5. There were no cytotoxic effects on Caco-2 cells, since there was no difference between the control cells and those treated with samples at various concentrations and time (*p* > 0.05). The result suggested that hydrolysate could be biocompatible and harmless to epithelial cell lines. Therefore, it could be applied for absorption and transportation studies in Caco-2 cell monolayer.

### 3.4. Transport Study of Digested Proteins

#### 3.4.1. Percentage of Permeability or Bioavailability

Transports of RKB, TBS and EWP hydrolysates at different concentrations (1 and 5 mg/mL) from apical (AP) and their corresponding Caco-2 cell monolayer permeates in basolateral (BL) side collected at different times (1, 2 and 4 h) are depicted in Figure 6. The percentage of permeability of RKB, TBS and EWP hydrolysates mainly ranged between 18% and 105%. There are various pathways reported for the transepithelial transport of peptides via human intestinal epithelium, including the proton-coupled peptide transporter 1 (PepT1)-mediated transport pathway, the tight junctions (TJs)-mediated paracellular pathway, and the vesicle-mediated transcytosis pathway [33]. In all the samples, varying permeability was observed, depending on time and digest concentrations used. The difference in permeation was probably related to the difference in size of the peptides generated during hydrolysis. In general, the enzyme specificity is greatly dependent on the type of substrate and it also leads to generation of peptides with different MWs. High permeation was noticed in TBS hydrolysate, compared to RKB and EWP hydrolysates, regardless of time and digest concentration. The TBS sample was extensively degraded by TPP and TLB at a level of 50% and several low MW peptides were generated, compared to the RKB and EWP samples (Figure 4). It is generally known that PepT1 is responsible for transporting dipeptides and some tripeptides [11]. Oligopeptides containing more than three AAs were able to pass across the Caco-2 cell monolayer via the TJs-mediated paracellular pathway [34]. Transcytosis was assumed to be the primary method of transport for peptides with more than 10 AAs [35]. TJs include a large number of tiny holes with a radius below 15 Å, which preferentially permit water-soluble low MW molecules to transport across [36]. Shimizu et al. [37] demonstrated that the permeability and transepithelial transportation of collagen peptides via the paracellular pathway are influenced by their MW.

The percentage of permeation of all samples augmented with increasing digest concentration and time (*p* < 0.05) (Figure 6). Similar results were seen for Gln-Ile-Gly-Leu-Phe (QIGLF), an ACE-inhibitory peptide, where higher QIGLF passed through Caco-2 cell monolayer with increasing time [34]. At 4 h, TBS hydrolysate with a concentration of 5 mg/mL had the highest permeation (105%), compared to RKB and EWP hydrolysates (*p* < 0.05). The majority of peptide degradation occurs by intracellular enzymes at the brush border membrane [38]. Oligopeptides are more likely hydrolyzed before and during the absorption process by different proteases and brush boundary membrane peptidases than dipeptides. TBS hydrolysate was further hydrolyzed by membrane peptidases, resulting in the increased AA and peptide contents. Generally, the peptide permeability via Caco-2 cell monolayer can be affected by several factors, including peptide determination methods, incubation time, initial concentration of peptide, Caco-2 cell monolayer TEER value, cell culture conditions, etc. [39]. Peptides from all samples were mostly absorbed and transported well across the Caco-2 cell monolayer. Therefore, permeation of samples through the Caco-2 cell monolayer could be influenced by the transportation time and digest concentrations.

#### 3.4.2. Size Distribution

MWs of peptides in RKB, TBS and EWP hydrolysates from the AP-side and their corresponding Caco-2 cells permeates collected from the BL-side are shown in Figure 7. The spectra showed several peaks, indicating the presence of several peptides in the samples with various MWs. Peptides generated after simulated *in vitro* digestion of RKB, TBS and EWP samples had total peak numbers of 30, 16 and 15 in the AP-side, respectively. The dominant peak in RKB, TBS and EWP hydrolysates represented 3068 Da, 2979 Da and 2176 Da MW peptides, respectively. TPP and TLB in a simulated *in vitro* digestion could generate peptides with different MWs for all the samples. The results coincided with varying DHs and protein patterns, in which cleavage of several peptide bonds were noticed and various low MW peptides were detected after protein hydrolysis by TPP and TLB at a level of 50% (Table 1 and Figure 4). The MW significantly affects the biological and functional properties of hydrolysates [40]. Many studies demonstrated that low MW peptides exhibited significant antioxidant, anti-tumor and ACE inhibitory activities [18,41,42,43].

Changes in the number of peaks or size distributions of peptides were observed in all samples after passing through the Caco-2 cell monolayer. The products of GI digestion interact with the intestinal lining cells and can pass through [44]. In the BL-side, RKB had a total peak number of 25, followed by EWP (22) and TBS (19). The total number of peaks for EWP and TBS were increased when hydrolysates passed through the Caco-2 cell monolayer. Some modifications might be caused by interactions with the intestinal cells and could be associated with brush border peptidase action. More or less 12 evolutionarily preserved peptidases are found in the intestinal microvilli of epithelial enterocytes, and they play a crucial physiological function in the cleavage of peptides into AA and small oligopeptides [44]. The results suggested that peptidases resulted in further degradation of peptides and generation of peptides with low MW. On the other hand, the total number of peaks for RKB were decreased after hydrolysates passed through the Caco-2 cell monolayer. On the BL-side of RKB, 16 peptides with the same MW were found as compared to the AP-side, while more or less peak intensity was noticeable. Those peptides were more likely withstood against intestinal peptidases action. As a consequence, peptides from the AP-side were less digested by peptidase of the Caco-2 cell monolayer and were able to pass across the Caco-2 cell monolayer. The results indicated that some peptides in RKB hydrolysate were degraded with peptidase, while others were resisted. Overall, TPP and TLB efficiently digested all proteins and produced various low MW peptides. Moreover, some of those peptides were further degraded by peptidase during transportation through the Caco-2 cell monolayer and produced smaller peptides or AAs that could be easily absorbed into the body. Therefore, size distribution of peptides could be influenced by enzyme action in both the GI tract and Caco-2 cell monolayer.

#### 3.4.3. Antioxidant Assay

Antioxidant activities of RKB, TBS and EWP hydrolysates from the AP-side and its corresponding Caco-2 cell monolayer permeates collected from the BL-side are shown in Figure 8. Hydrolysates generated after simulated *in vitro* digestion of the RKB sample showed the highest ABTS (3.16 mmol TE/L) and DPPH (3.87 mmol TE/L) radical-scavenging activities (RSA) in the AP-side, compared to other samples (*p* < 0.05). Conversely, the lowest ABTS-RSA (1.69 mmol TE/L) and DPPH-RSA (2.40 mmol TE/L) were observed in EWP and TBS, respectively. The results suggested that TPP and TLB with different enzyme specificities for varying substate generated short-chain peptides with different MWs, which could scavenge ABTS and DPPH radicals differently. In general, peptides or proteins found in hydrolysates act as a hydrogen donor that may interact with radicals to change them into more stable products, thus stopping the radical chain reaction [45]. Additionally, significant factors in the radical scavenging activity include peptide size, AA sequence, and peptide conformation [46]. Moreover, plant proteins may contain some phytochemicals such as, phenolic compounds, which can serve as free radical scavengers [47].

Changes in antioxidant activities for all samples were observed in permeates after transepithelial transport. When the hydrolysate passed through the Caco-2 cell monolayer, ABTS-RSA and DPPH-RSA were lower in all hydrolysate samples. Similar results were reported by Zhang, Tong, Qi, Wang, Li, Sui and Jiang [16], in which the antioxidant activities of the permeates were changed to some extent after Caco-2 cell absorption. The results suggested that modifications or degradation of some peptides occurred during the absorption and transportation through the Caco-2 cell monolayer, as confirmed by changes in MW distribution. As a result, the antioxidant activities of all samples were decreased to certain extent. Among all samples, RKB permeates showed the highest ABTS-RSA (2.01 mmol TE/L) and DPPH-RSA (2.49 mmol TE/L) after being transported across the Caco-2 cell monolayer (*p* < 0.05). Some peptides with antioxidant activities were plausibly transported across the monolayer, indicating the retained antioxidant activities. The obtained results were in tandem with MW distribution, in which a higher number of peptides from RKB hydrolysate were transported across the monolayer. Therefore, antioxidant activities of peptides derived from the simulated *in vitro* digestion of RKB, TBS and EWP were influenced by the Caco-2 cell monolayer. Permeates partially maintained antioxidant activities and, thus, could reduce the risk of degenerative diseases [48,49].

## 4. Conclusions

Hard capsules containing TPP and TLB in blister packs were quite stable over prolonged storage times, particularly at low temperatures (4 °C), with some losses in activity when stored for a longer time. Moreover, TPP and TLB hydrolyzed RKB, TBS and EWP in a simulated *in vitro* digestion and produced different peptides with varying MWs. The generated peptides showed antioxidant activity and were partially maintained after transepithelial transport. Therefore, TPP and TLB could be used as supplements for people lacking in digestive enzymes without religious limitations. Additionally, food-derived bioactive peptides generated after GI digestion possessed some potential antioxidant activity, which could assist in improving human health.

## Figures and Tables

**Figure 1 foods-12-00125-f001:**
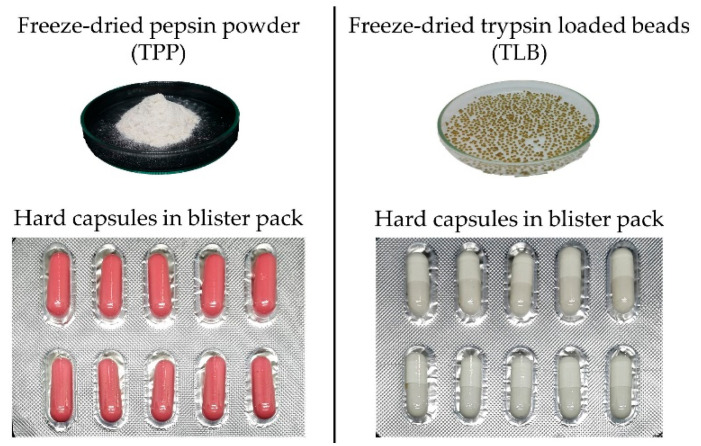
Freeze-dried pepsin powder (TPP) and trypsin loaded beads (TLB) filled into hard capsules and packed in blister packs.

**Figure 2 foods-12-00125-f002:**
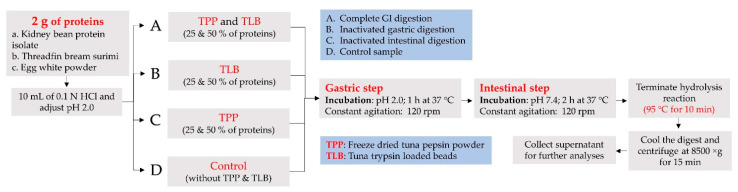
Flow diagram of simulated *in vitro* gastrointestinal (GI) digestion of some proteins by freeze-dried tuna pepsin powder (TPP) and trypsin loaded beads (TLB).

**Figure 3 foods-12-00125-f003:**
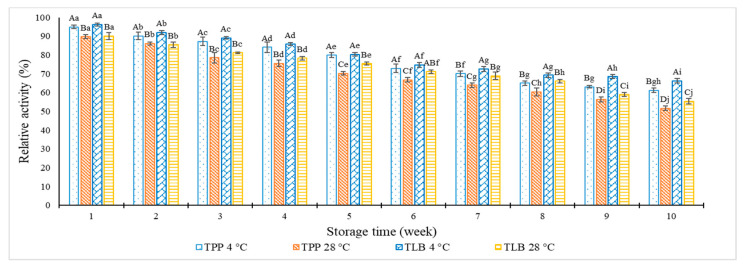
Relative activity of tuna pepsin powder (TPP) and trypsin loaded beads (TLB) in blister pack during 10 weeks of storage at different temperatures (4 °C and 28 °C). Different lowercase letters on the bars within the same storage time denote significant differences (*p* < 0.05). Different uppercase letters on the bars within the same storage temperature denote significant differences (*p* < 0.05). Error bars represent standard deviation (*n* = 3).

**Figure 4 foods-12-00125-f004:**
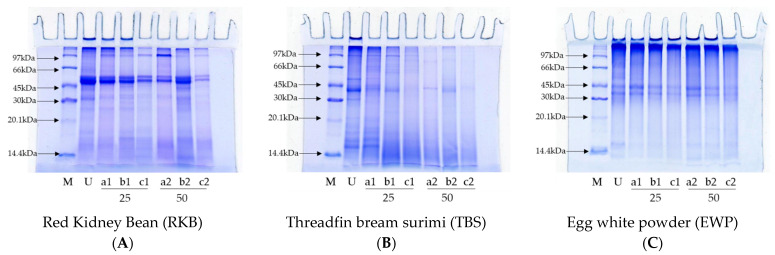
Protein patterns of (**A**) red kidney bean (RKB); (**B**) threadfin bream surimi (TBS); and (**C**) egg white powder (EWP) without digestion (U) and subjected to digestion under different simulated *in vitro* gastrointestinal (GI) tract conditions without and with tuna pepsin powder (TPP) and/or trypsin-loaded beads (TLB) at different levels. a: In the absence of TPP b: In the absence of TLB and c: In the presence of both TPP and TLB. Number denotes the level of enzymes (%, based on protein content). M: Low molecular weight marker.

**Figure 5 foods-12-00125-f005:**
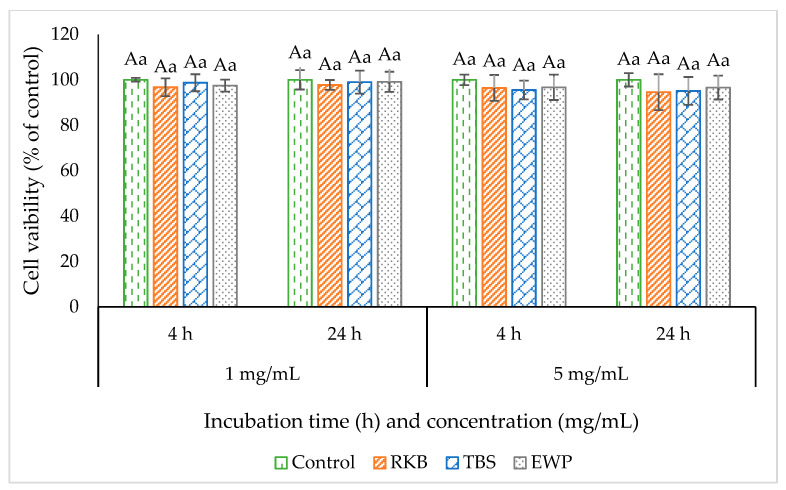
Effect of red kidney bean protein (RKB)-, threadfin bream surimi (TBS)- and egg white protein (EWP)-digested samples on the cell viability of Caco-2 cells (% of control). Different uppercase letters on the bars within the same incubation time denote significant differences (*p* < 0.05). Different lowercase letters on the bars within the same concentration denote significant differences (*p* < 0.05). Error bars represent standard deviation (*n* = 3).

**Figure 6 foods-12-00125-f006:**
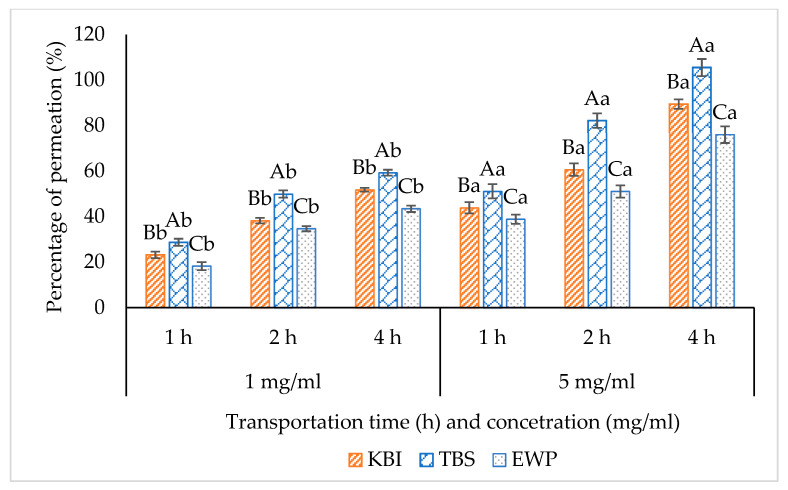
Percentage of permeation of red kidney bean protein (RKB), threadfin bream surimi (TBS) and egg white protein (EWP) at different concentration and its corresponding Caco-2 cell monolayer permeates collected from basolateral at different transportation time. Different uppercase letters on the bars within the same transportation time and concentration denote significant differences between the samples (*p* < 0.05). Different lowercase letters on the bars within the same transportation time denote significant differences between different concentrations (*p* < 0.05). Error bars represent standard deviation (*n* = 3).

**Figure 7 foods-12-00125-f007:**
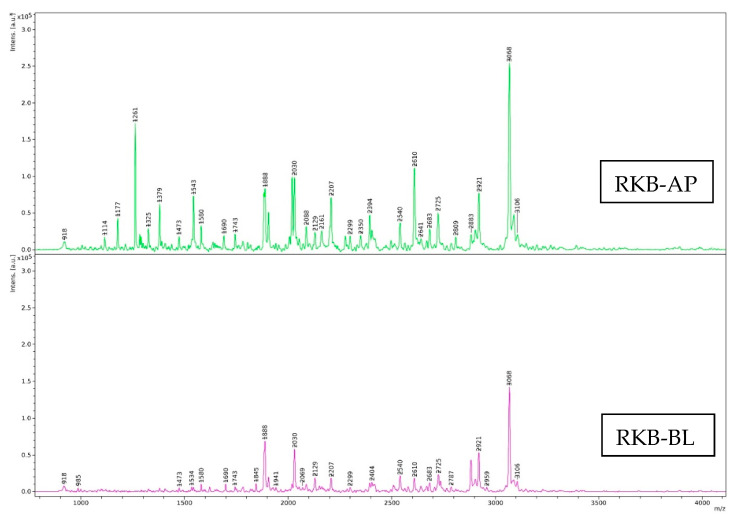
Size distribution of peptides in hydrolyzed proteins including red kidney bean protein (RKB), threadfin bream surimi (TBS) and egg white protein (EWP) before transport (AP) and after 4-h transport appeared as basolateral (BL)-side permeates.

**Figure 8 foods-12-00125-f008:**
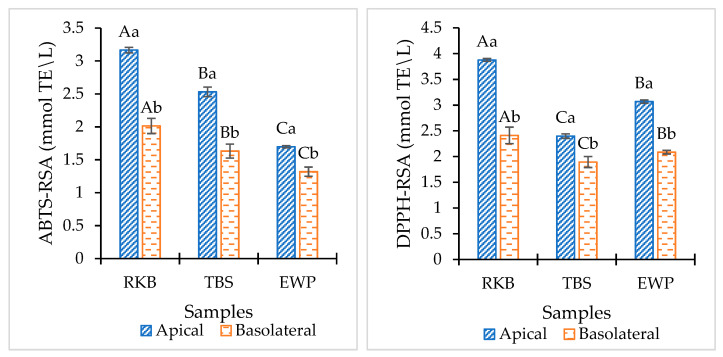
Antioxidant activity of red kidney bean isolate (RKB), threadfin bream surimi (TBS) and egg white powder (EWP) hydrolysate before transport (Apical) and after 4-h transport appeared in basolateral-side permeates (Basolateral). Different uppercase letters on the bars within the same side of Caco-2 cell monolayer denote significant differences between the samples (*p* < 0.05). Different lowercase letters on the bars within the same protein used denote significant differences between apical and basolateral side (*p* < 0.05). Error bars represent standard deviation (*n* = 3).

**Table 1 foods-12-00125-t001:** Degree of hydrolysis of red kidney bean (RKB), threadfin bream surimi (TBS) and egg white protein (EWP) in various simulated *in vitro* gastrointestinal (GI) model systems without and with fish pepsin and/or trypsin at different levels.

Treatments	Enzyme Level (*w*/*w*, Based on Protein)	Degree of Hydrolysis (%)		
		Red Kidney Bean Isolate (RKB)	Threadfin Bream Surimi (TBS)	Egg White Proteins (EWP)
Absence of TPP	25	24.79 ± 0.64 cF	32.98 ± 1.65 aF	28.82 ± 1.49 bF
	50	48.36 ± 1.77 cD	60.04 ± 1.10 aD	51.64 ± 1.04 bD
Absence of TLB	25	29.15 ± 1.65 cE	36.75 ± 0.14 aE	31.37 ± 0.57 bE
	50	55.41 ± 1.30 cC	66.94 ± 1.08 aC	59.98 ± 0.30 bC
Presence of both TPP and TLB	25	68.27 ± 1.53 cB	81.41 ± 1.17 aB	70.86 ± 1.41 bB
	50	76.82 ± 0.82 cA	89.13 ± 1.15 aA	74.16 ± 0.99 bA

TPP: Tuna pepsin powder. TLB: Trypsin-loaded beads. Values are presented as mean ± SD (*n* = 3). Different uppercase letters in the same column indicate significant difference (*p* < 0.05). Different lowercase letters in the same row indicate significant difference (*p* < 0.05).

## Data Availability

The data presented in this study are available in article.

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
