# Peer review of "Use of Tuna Visceral Pepsin in Combination with Trypsin as Digestion Aid: Enhanced Protein Hydrolysis and Bioavailability"

_foods, 2022, doi:10.3390/foods12010125_

Round 1

Reviewer 1 Report

My comments:

1. It would be easier for the reader the presence of a flow diagramm in chapter 2.5

2. It would of importance to see photographs of the cell cultures before and after various treatments

Author Response

Responses to reviewer

Reviewer: 1

***** We are thankful to the reviewer for understanding our work. We have considered the suggestion made by the reviewer and have accordingly corrected it in the revised text.

  1. It would be easier for the reader the presence of a flow diagram in chapter 2.5

***** Thank you for the suggestion. A flow diagram has been added. Please see Figure 2.

  1. It would of importance to see photographs of the cell cultures before and after various treatments

***** Thank you for the reviewer’s suggestion. However, the cell photograph (morphology) observation was out of this study's scope. Caco-2 monolayer in transwell plate was developed with the purpose to mimic the human gastrointestinal tract used for peptide absorption study. The integrity of the monolayer cell was monitored by the electrical resistance system (using TEER) before and after transportation study to confirm the good health and integrity of the cell. Therefore, this model was used only for peptide absorption study, and the cell morphology was not necessary.

Reviewer 2 Report

Protease plays an important role in protein hydrolysis. The paper authored by Patil et al., showed that freeze-dried tuna pepsin powder (TPP) was prepared with the aid of maltodextrin (10%) and trehalose (5%) to Enhance protein hydrolysis and bioavailability. The enzyme was encapsulated in chitosan/alginate gel to prepare trypsin loaded pellets (TLB) with 5% 42 glycerol. The hard capsules of TPP and TLB were prepared, and the effects of storage temperature and time on enzyme activity were studied. The effects of TPP and TLB on the hydrolysis of different proteins (RKB, TBS and EWP) in intestinal digestion were simulated in vitro. Through research and analysis, it was found that the peptides produced by hydrolysis could be transported through the monolayer of Caco-2 cells, and the peptides produced had antioxidant activity, which was partially maintained after transport. Therefore, TPP and TLB can be used as a supplement for people who lack digestive enzymes, and they can replace the commercial cow and pig pepsin restricted by religion. In addition, the food derived bioactive peptides obtained from gastrointestinal digestion have certain antioxidant activity, which can help improve human health.

Major comment

1. The research content of Caco-2 cell composition article needs to be introduced in detail in introduction.

2. DH testing methods need to be elaborated in the article.

3. The article points out that tuna visceral pepsin and trypsin can replace commercially available pig or cow pepsin and trypsin with religious restrictions, but it does not compare tuna visceral pepsin and trypsin with pig or cow pepsin and trypsin. Besides the effect of hydrolysis, whether other aspects are suitable for replacement.

Minor comment

1. It is necessary to describe in detail the reasons for carrying out the experiment of putting enzymes into hard capsules to observe the effect of storage temperature and time on enzyme activity.

2. References should be added to Line484.

3. There is some confusion in the significance labels in Figures 5 and 7, which is not easy to understand.

Author Response

Responses to reviewer

Reviewer: 2

Protease plays an important role in protein hydrolysis. The paper authored by Patil et al., showed that freeze-dried tuna pepsin powder (TPP) was prepared with the aid of maltodextrin (10%) and trehalose (5%) to enhance protein hydrolysis and bioavailability. The enzyme was encapsulated in chitosan/alginate gel to prepare trypsin loaded pellets (TLB) with 5% glycerol. The hard capsules of TPP and TLB were prepared, and the effects of storage temperature and time on enzyme activity were studied. The effects of TPP and TLB on the hydrolysis of different proteins (RKB, TBS and EWP) in intestinal digestion were simulated in vitro. Through research and analysis, it was found that the peptides produced by hydrolysis could be transported through the monolayer of Caco-2 cells, and the peptides produced had antioxidant activity, which was partially maintained after transport. Therefore, TPP and TLB can be used as a supplement for people who lack digestive enzymes, and they can replace the commercial cow and pig pepsin restricted by religion. In addition, the food derived bioactive peptides obtained from gastrointestinal digestion have certain antioxidant activity, which can help improve human health.

***** We greatly appreciate the reviewer for the valuable time spent on the comments and suggestions to improve our manuscript. All queries have been responded and the corrections have been made in the revised manuscript. 

Major comment

  1. The research content of Caco-2 cell composition article needs to be introduced in detail in introduction.

***** Thank you for the comment. The additional details of the Caco-2 cell have been added in the introduction with related references as follows:

Generally, Caco-2 monolayer has been widely used as an in vitro model for intestinal absorption across the epithelium. Although this type of cell is a human colorectal cancer cell line, it displays an enterocyte-like phenotype after culturing as a monolayer for 20-22 days (1). Many types of enzymes and transport molecules expressed in the human small intestine were also found in fully differentiated Caco-2 monolayer cell (2). Moreover, the previous study demonstrated the correlation between this model and in vivo human absorption for a wide range of drug molecules (3).

Please see line number 49-55.

  1. DH testing methods need to be elaborated in the article.

***** Thank you for the suggestion. Details for the method used for measuring the degree of hydrolysis (DH) of the digest and the equation used have been added in the revised text. Please see line number 149-161.

  1. The article points out that tuna visceral pepsin and trypsin can replace commercially available pig or cow pepsin and trypsin with religious restrictions, but it does not compare tuna visceral pepsin and trypsin with pig or cow pepsin and trypsin. Besides the effect of hydrolysis, whether other aspects are suitable for replacement.

***** Thank you for the point raised by the reviewer. In our previously published study, we found that tuna pepsin had comparable hydrolysis toward several proteins to commercial pepsin, especially at a higher levels (15 units/g protein). However, we did not compare tuna visceral pepsin and trypsin with pig or cow pepsin and trypsin in this study. Therefore, the sentences regarding the fish enzymes replacement with commercial enzymes have been deleted to avoid confusion.

Please see line numbers 300-301, 515.

Minor comment

  1. It is necessary to describe in detail the reasons for carrying out the experiment of putting enzymes into hard capsules to observe the effect of storage temperature and time on enzyme activity.

***** Thank you for your suggestion. The storage condition and packaging might cause degradation or autolysis of enzymes that lead to loss of enzyme activity. Therefore, enzymes were packed in hard capsules to minimise the loss of enzyme activity and to commercialize fish enzymes. More importantly, authors packed the enzymes, both pepsin powder and trypsin bead, in hard capsule to mimic the real product available in the market, in which the extract, compound or enzyme is commonly packed in hard capsule due to the ease of operation and low cost. The information obtained from this study will be of benefit to enzyme supplementation manufacturing as well as the fish processing industry.

The objective of the experiment has been added to the methodology in the revised manuscript. Please see line numbers 109-112.

  1. References should be added to Line 484.

***** Thank you for the suggestion. The references have been added to the revised manuscript. Please see line number 500.

  1. There is some confusion in the significance labels in Figures 5 and 7, which is not easy to understand.

***** Thank you for your comment. The corrections have been made. Please see line numbers 421-423, and 504-506.

Round 2

Reviewer 2 Report

The author answered all the questions of the reviewers within a basically acceptable range.

Author Response

Response to reviewer

Reviewer: 2

The author answered all the questions of the reviewers within a basically acceptable range.

***** We are thankful to the reviewer for the valuable time spent on the comments and suggestions to improve our manuscript. Thank you very much.
